

# Quality improvement bundles to decrease hypothermia in very low/extremely low birth weight infants at birth: a systematic review and meta-analysis

Guichao Zhong[1], Jie Qi[1], Lijuan Sheng[1], Jing Zhuang[1], Zhangbin Yu[1] and Benqing Wu[2]

[1] Department of Neonatology, Shenzhen People's Hospital (The Second Clinical Medical College, Jinan University; The First Affiliated Hospital, Southern University of Science and Technology), Shenzhen, Guangdong, China

[2] Department of Neonatology, Shenzhen Guangming District People's Hospital, Shenzhen, Guangdong, China

Corresponding authors
Zhangbin Yu, yuzhangbin@126.com
Benqing Wu,
wubenqing783@126.com

## ABSTRACT

**Background:** Numerous studies have demonstrated that hypothermia in preterm infants correlates with increased morbidity and mortality, especially among those with very low or extremely low birth weights (VLBW/ELBW). An increasing number of healthcare facilities are implementing quality improvement (QI) bundles to lower the incidence of hypothermia at birth in this vulnerable population. However, the effectiveness and safety of these interventions have yet to be fully assessed. A meta-analysis is necessary to evaluate the efficacy and safety of QI bundles in reducing hypothermia at birth among VLBW/ELBW infants.

**Methods:** We searched PubMed, Embase, the Cochrane Library and Web of Science through April 22nd, 2024. Study selection, data extraction, quality evaluation and risk bias assessment were performed independently by two investigators. Meta-analysis was performed using Review Manager 5.4.1.

**Results:** A total of 18 studies were included for qualitative analysis and 12 for meta-analysis. For VLBW infants, meta-analysis revealed a reduction in hypothermia and an increase in hyperthermia following the introduction of QI bundles (mild hypothermia, OR 0.22, 95% CI [0.13–0.37]; moderate hypothermia, OR 0.18, 95% CI [0.15–0.22]; hyperthermia, OR 2.79, 95% CI [1.53–5.09]). For ELBW infants, meta-analysis showed a decrease in hypothermia but no increase in hyperthermia after implementing QI bundles (mild hypothermia, OR 0.46, 95% CI [0.26–0.81]; moderate hypothermia, OR 0.21, 95% CI [0.08–0.58]; hyperthermia, OR 1.10, 95% CI [0.22–5.43]).

**Conclusion:** QI bundles effectively reduce hypothermia in VLBW/ELBW infants, but they may also increase hyperthermia, especially in VLBW infants.

# INTRODUCTION

The World Health Organization (WHO) defines the range of 36.5 degrees Celsius (°C) to 37.5 °C as the normal body temperature for newborns. Body temperatures of 36.0–36.4 °C are defined as mild hypothermia and 32.0–35.9 °C as moderate hypothermia (*World Health Organization MNHS & Motherhood, 1997*). Neonatal hypothermia significantly contributes to both neonatal mortality and morbidities, such as hypoglycemia, necrotizing enterocolitis, intraventricular hemorrhage, and late-onset sepsis, particularly among very low/extremely low birth weight (VLBW/ELBW) infants (*Cordeiro et al., 2021*). This vulnerability stems from their large surface area relative to body mass, limited subcutaneous fat, and immature thermoregulatory mechanisms (*Perlman & Kjaer, 2016*). Studies have reported a wide variation in hypothermia rates among VLBW infants, ranging from 31% to 78% (*Bhatt et al., 2007*). Additionally, for every 1 °C decrease in admission temperature, the odds of late-onset sepsis increase by 11%, and the odds of mortality increase by 28% (*Laptook, Salhab & Bhaskar, 2007*). Given these potential adverse outcomes, preventing hypothermia in VLBW and ELBW infants is of paramount importance. A systematic review has evaluated the risk factors for hypothermia in this vulnerable population (*Shi et al., 2023*), and while single interventions such as skin-to-skin care, thermal mattresses, or plastic wraps have proven effective in improving temperature in randomized controlled trials (*McCall et al., 2018*), they remain insufficient in fully preventing hypothermia in VLBW/ELBW infants. Quality improvement (QI) in health care is defined by the Agency for Healthcare Research and Quality (AHRQ) as "the framework to systematically improve the ways care is delivered to patients." Various QI methodologies have been widely used to identify and close the gap between desired and actual results in health care (*Valeras, 2019*). These methodologies have also been extensively applied in the field of neonatology (*Norman et al., 2019*; *Pearlman, 2022*). To reduce the incidence of hypothermia in VLBW/ELBW infants, efforts have been initiated to implement various quality improvement measures, collectively referred to as quality improvement bundles (QI bundles). These bundles comprise evidence-based interventions aimed at maintaining normothermia from birth through transportation to the neonatal intensive care unit (NICU). Common interventions range from environmental temperature control and the use of polyethylene wraps or caps to the use of radiant warmers or incubators, as well as establishing multidisciplinary teams and training their members. While the implementation of these QI bundles in reducing hypothermia has been endorsed by the American Heart Association (AHA) and the American Academy of Pediatrics (AAP) in the 2020 neonatal resuscitation guidelines (*Aziz et al., 2021*), variability in practices across different settings persists, and there is a lack of consensus on the most effective components. A systematic review (*Donnellan et al., 2020*) evaluated the impact of quality improvement initiatives (QIs) on the admission temperatures of premature/VLBW infants in neonatal intensive care units (NICUs). The review included 10 QI studies, which demonstrated that the implementation of thermoregulation QIs can positively impact admission temperatures for premature or VLBW infants admitted to the
NICU. However, this study did not perform a quantitative data synthesis (meta-analysis) and did not analyze the effectiveness of QI measures in reducing mild hypothermia *vs.* moderate hypothermia. Additionally, the study did not separately analyze the effectiveness of QI measures in preventing hypothermia in VLBW infants and ELBW infants. On the other hand, Some other systematic reviews and meta-analyses aimed at improving neonatal hypothermia (*Abiramalatha et al., 2021*; *McCall et al., 2018*; *Ramaswamy et al., 2023*) primarily include randomized controlled trials (RCTs) and quasi-randomized controlled trials, rather than quality improvement studies. In addition, these systematic reviews and meta-analyses did not specifically focus on VLBW/ELBW infants, who are at the highest risk for hypothermia, but rather included a broader population with a wider range of gestational ages. Therefore, we aimed to systematically review these QI studies and synthesize the data to assess the effectiveness and safety of QI bundles in preventing hypothermia in VLBW/ELBW infants at birth.

# MATERIALS AND METHODS

This study was conducted following the PRISMA guidelines (*Liberati et al., 2009*). The protocol for this systematic review and meta-analysis was registered on PROSPERO under the accession number CRD42024534226 and can be accessed at https://www.crd.york.ac.uk/prospero/#searchadvanced. Ethical approval was not sought as this study involved secondary analysis of published data.

## Data sources and search strategy

A comprehensive literature search was conducted in electronic databases including PubMed, Embase, the Cochrane Library and Web of Science. The final search was completed on April 22nd, 2024, and no search limits or restrictions were applied initially to broaden the search scope. Appendix 1 describes the detailed search strategy across individual databases.

## Inclusion and exclusion criteria

The criteria for including studies in this systematic review and meta-analysis are built on the PICOS framework: (1) Population (P), VLBW/ELBW infants, or preterm infants with a gestational age (GA) of 32 weeks or less. (2) Intervention (I), application of QI bundles aimed at preventing hypothermia at birth. (3) Comparison (C), VLBW/ELBW infants before the implementation of QI bundles as the comparison. (4) Outcome (O), the incidence of hypothermia from birth to transfer to the NICU. (5) Study design (S), the eligible study design is that of a Quality Improvement Project (QIP) that applies QI bundles. Exclusion criteria were as follows: (1) Participants being newborns with a gestational age over 32 weeks and a birth weight exceeding 1,500 g. (2) Studies not related to quality improvement, such as randomized or quasi-randomized trials of single or multiple interventions. (3) Temperature measurement is not between birth and admission to the NICU or the site of temperature measurement is not clear. (4) Studies only available as conference abstracts.
## Study selection and data extraction

Two independent reviewers (GZ and JQ) screened titles and abstracts of identified studies for relevance. Full-text articles were retrieved for potentially relevant studies and assessed for eligibility based on predefined inclusion and exclusion criteria. Discrepancies between reviewers were resolved through discussion, with involvement of a third reviewer (BW) if consensus could not be reached. Two authors (GZ and JQ) independently extracted data using a standardized data collection form. The extracted information included study author, year of publication, location, study duration, sample size, participant demographics, site of temperature measurement, outcomes (incidence of mild hypothermia, moderate hypothermia, and hyperthermia) and components of the QI bundles. Disagreements were resolved through discussion or consultation with a third author (ZY) if necessary.

## Quality evaluation and risk bias assessment

Two investigators (GZ and JQ) independently assessed the quality of the selected studies using the Quality Improvement Minimum Quality Criteria Set (QI-MQCS) (*Hempel et al., 2015*). This set encompasses 16 content categories: Organizational Motivation, Intervention Rationale, Intervention Description, Organizational Characteristics, Implementation, Study Design, Comparator, Data Source, Timing, Adherence/Fidelity, Health Outcomes, Organizational Readiness, Penetration/Reach, Sustainability, Spread, and Limitations. Each study was evaluated across these domains, earning 1 point for meeting the minimum criteria in each domain, and 0 points otherwise. The studies were subsequently classified based on their quality: scores above 10 signified high quality, scores between 7 and 10 indicated medium quality, and scores below 7 denoted low quality. Any discrepancies were resolved through group consensus.

## Statistical analysis

When conducting the synthesis of effect sizes, data from VLBW infants and ELBW infants was categorized separately according to mild hypothermia, moderate hypothermia, and hyperthermia. Only studies included in the systematic review that provide binary variable data (or derived binary variables) for hypothermia or hyperthermia were included in the meta-analysis. If a study provides data for multiple groups, it will be separately included in the respective groups for data synthesis. We summarized the outcomes by reporting the odds ratio (OR) and 95% confidence interval (CI), and presented them using forest plots. Meta-analyses were performed with Review Manager 5.4.1 software. We assessed the heterogeneity of the studies using $I^2$ statistics. A random-effects model was employed to address substantial heterogeneity between studies ($I^2 \geq 50\%$), defaulting to a fixed-effects model when heterogeneity was absent ($I^2 < 50\%$). Sources of heterogeneity were investigated, and sensitivity analyses were conducted to determine the impact of individual study results on the overall effect size. $P < 0.05$ was considered significant. When the number of studies included in the meta-analysis exceeds ten, a funnel plot generated by Review Manager 5.4.1 software will be used to assess publication bias.

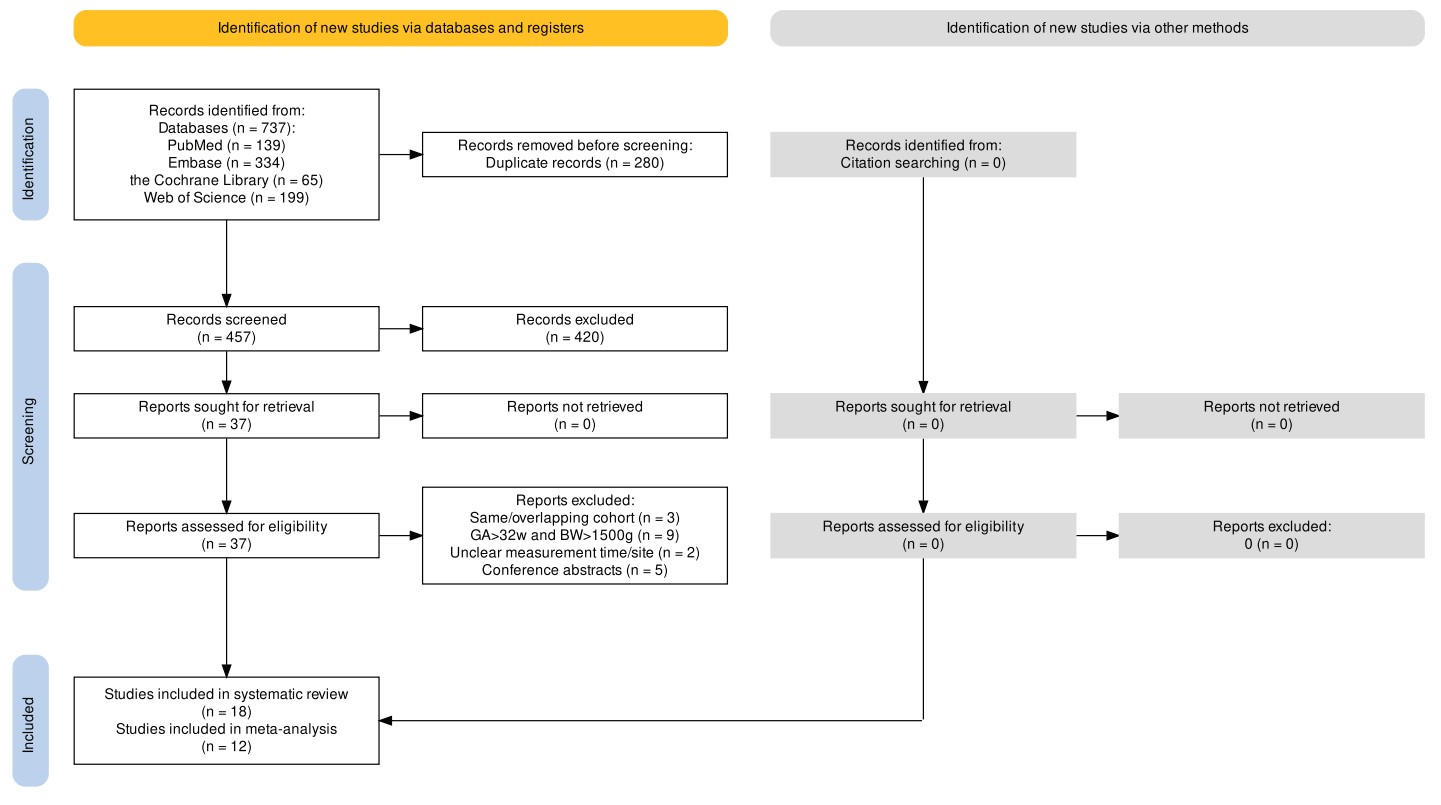

**Figure 1  PRISMA flowchart summarizing the article selection process.**

## RESULTS

### Study selection

A total of 737 articles were retrieved from the electronic databases including PubMed (139), Embase (334), the Cochrane Library (65), and Web of Science (199). After removing duplicates, 457 articles remained and were screened for eligibility based on title and abstract. Out of the screened articles, 37 were selected for full-text screening to assess their suitability for inclusion. Among these, 19 articles were excluded due to reasons such as having the same or overlapping study cohort (three) (*Frazer et al., 2018*; *Harriman et al., 2018*; *Wang et al., 2022*), including newborns with a birth weight >1,500 g and GA > 32 weeks (nine) (*Andrews et al., 2018*; *Choi et al., 2018*; *Harer et al., 2017*; *Howes & Keir, 2018*; *Patodia et al., 2021*; *Pratik et al., 2023*; *Ramjist et al., 2024*; *Shaw et al., 2018*; *Sprecher et al., 2021*), unclear temperature measure time/site (two) (*Glenn et al., 2021*; *Keir & Cavallaro, 2021*), and only conference abstracts are available (five) (*Bobby et al., 2014*; *Dale et al., 2021*; *Shaw et al., 2016*; *Thornton, Voos & McNellis, 2014*; *Woolley et al., 2019*). Following the full-text screening, 18 articles met the criteria for qualitative synthesis. Among these, 12 articles were deemed suitable for quantitative synthesis through meta-analysis. Specifically, for the meta-analysis, for VLBW infants, four, eight, and six studies were included for mild hypothermia, moderate hypothermia, and hyperthermia, respectively; for ELBW infants, two, three, and three studies were included for mild hypothermia, moderate hypothermia,

**Table 1  Characteristics of included studies.**

| References | Location | Duration | Size | GA/BW | Site of temperature measurement | Outcomes |
|---|---|---|---|---|---|---|
| *Kent & Williams (2008)* | Australia | 2000–2005 | 156 | GA < 32 w | Axilla | (1) (2) (4) |
| *Lee, Ho & Rhine (2008)* | USA | 2003–2006 | 304 | BW < 1,500 g | Rectum | (3) |
| *Billimoria et al. (2013)* | USA | 2005–2010 | 209 | BW < 1,000 g | Axilla | (2) (3) |
| *DeMauro et al. (2013)* | USA | 2009–2012 | 160 | BW < 1,250 g | Axilla | (1) (2) (4) |
| *Godfrey et al. (2013)* | USA | 2010–2011 | 81 | GA < 28 w | Rectum | (1) (2) (4) |
| *Manani et al. (2013)* | USA | 2006–2011 | 289 | BW < 1,500 g | Axilla | (2) (3) |
| *Castrodale & Rinehart (2014)* | USA | 2008–2011 | 225 | GA < 28 w | Axilla | (4) |
| *Pinheiro et al. (2014)* | USA | 2007–2012 | 666 | BW < 1,500 g | Axilla | (3) (5) |
| *Sivanaridan, Sankar & Deorari (2016)* | India | 2015–2016 | 56 | GA < 32 w | Axilla | (3) |
| *Yip et al. (2017)* | Singapore | 2007–2013 | 1,199 | BW < 1,500 g | Axilla/rectum | (1) (2) (4) |
| *Caldas et al. (2018)* | Brazil | 2012–2016 | 475 | BW < 1,500 g | Axilla | (2) (3) |
| *Bhatt et al. (2020)* | USA | 2014–2019 | 200 | BW < 1,000 g | Axilla | (1) (2) (4) |
| *Croop et al. (2020)* | USA | 2012–2017 | 214 | GA < 27 w | Axilla | (4) |
| *Dixon et al. (2021)* | USA | 2017–2019 | 67 | GA < 32 w | Axilla | (1) |
| *Frazer et al. (2021)* | USA | 2014–2017 | 334 | GA < 32 w/BW < 1,500 g | Axilla | (2) (3) |
| *Young et al. (2021)* | UK | 2011–2019 | 599 | GA < 30 w | Axilla | (1) (2) (4) |
| *Bi et al. (2022)* | China | 2018–2020 | 750 | BW < 1,500 g | Rectum | (1) (2) (3) (4) |
| *Singh et al. (2022)* | Australia | 2013–2017 | 568 | GA ≤ 32 w/BW < 1,500 g | Axilla | (1) (2) (3) (4) |

**Note:**
(1) Proportion of infants with a body temperature < 36.5 °C; (2) Proportion of infants with a body temperature > 37.5 °C;
(3) Proportion of infants with a body temperature < 36.0 °C; (4) Proportion of infants with a body temperature ranging from 36.5 °C to 37.5 °C; (5) Proportion of infants with a body temperature > 38.0 °C; GA, gestational age; BW, birth weight.

and hyperthermia, respectively. The detailed flowchart illustrating the screening and selection process is provided in Fig. 1, following the PRISMA guidelines.

## Study characteristics

The included 18 studies originated from diverse locations, with 11 from the USA (*Bhatt et al., 2020*; *Billimoria et al., 2013*; *Castrodale & Rinehart, 2014*; *Croop et al., 2020*; *DeMauro et al., 2013*; *Dixon et al., 2021*; *Frazer et al., 2021*; *Godfrey et al., 2013*; *Lee, Ho & Rhine, 2008*; *Manani et al., 2013*; *Pinheiro et al., 2014*), two from Australia (*Kent & Williams, 2008*; *Singh et al., 2022*), and one each from India (*Sivanaridan, Sankar & Deorari, 2016*), Singapore (*Yip et al., 2017*), Brazil (*Caldas et al., 2018*), the UK (*Young et al., 2021*) and China (*Bi et al., 2022*). The publication timeline of the selected studies spans from 2008 to 2022. Predominantly, the quality improvement initiatives (17/18) were conducted at a single center, and only one (*Bi et al., 2022*) was multicenter (5 NICUs). The sample sizes varied considerably, ranging from 59 to 1,199 (total = 6,552 infants). Table 1 displays the characteristics of the included studies.

## Quality and risk of bias assessment

The assessment of the included studies was conducted utilizing the Quality Improvement Minimum Quality Criteria Set (QI-MQCS). The results showed study scores ranging from 9 to 15, with five categorized as medium quality and 13 as high quality, detailed in

Appendix 2. Each study met the minimum quality requirements in seven of the 16 evaluated domains: organizational motivation, rationale behind the intervention, detailed description of the intervention, implementation strategies, description of the comparator, timing of the intervention, and the health outcomes observed. However, a predominant issue in most studies was the lack of detailed information on adherence to intervention protocols (adherence/fidelity: 14 out of 18 studies). Similarly, descriptions lacked details on the sustainability of interventions (sustainability: 12 out of 18 studies), potential for scalability or replication (spread: 12 out of 18 studies), and adequately outlining barriers and facilitators of organizational readiness for the interventions (organizational readiness: 11 out of 18 studies).

## Bundle components

A total of 10 components were identified across the 18 studies, with the number of components per bundle varying. The number of components ranged from four to nine. The most common components included formation of a multidisciplinary expert team, development of evidence-based interventions, education of hospital staff (18/18); using plastic wrap or bag (18/18); and use of radiant warmer or incubator (18/18). These were followed by keeping the head in plastic wrap, using polyurethane-lined hats, or stocking-knit caps (16/18); increasing ambient temperature (14/18); regular reinforcements or ongoing process review and feedback loop (12/18); using warming mattresses (10/18); monitoring temperatures (6/18); maintaining plastic bag/wrap integrity during the resuscitation process (2/18); and delaying anthropometric data collection and non-essential procedures (2/18). Table 2 displays the 10 components included in the QI bundles.

## Decrease in mild hypothermia (body temperature < 36.5 °C)

Among the 12 studies included in the meta-analysis, four provided dichotomous data on the incidence of mild hypothermia before and after QI in neonates with a birth weight of <1,500 g or GA < 32 weeks (*Bi et al., 2022*; *Dixon et al., 2021*; *Singh et al., 2022*; *Yip et al., 2017*). Westmead Hospital in Australia conducted a QI project over a 5-year period targeting VLBW infants and infants with a GA < 32 w (sample size: 568) (*Singh et al., 2022*). The project resulted in a significant reduction in the incidence of mild hypothermia, decreasing from 64.8% to 36.7%. KK Women's and Children's Hospital in Singapore also implemented a QI project for over 5 years with a sample size of 1,199 (including 168 before QI, 1,031 during and after QI) (*Yip et al., 2017*). The incidence of body temperature <36.5 °C decreased from 75.5% to 46.8%. A 36-month QI project conducted in Shandong, China, from January 2018 to December 2020, involved five NICUs and included a total sample size of 750 (*Bi et al., 2022*). The study observed a significant reduction in the incidence of mild hypothermia among VLBW infants, decreasing from 95.9% to 71.3%. Another QI project (sample size: 67) from Duke University School of Nursing (*Dixon et al., 2021*) reported that the incidence of mild hypothermia decreased from 33.3% (18/54) to 0% (0/13). The combined result showed that QI efforts substantially reduce the incidence of mild hypothermia in VLBW infants (OR 0.22, 95% CI [0.13–0.37]) (Fig. 2).

**Table 2  Interventions included in the QI bundles.**

| References | (1) | (2) | (3) | (4) | (5) | (6) | (7) | (8) | (9) | (10) |
|---|---|---|---|---|---|---|---|---|---|---|
| *Kent & Williams (2008)* | + | + | + | | + | | | | | |
| *Lee, Ho & Rhine (2008)* | + | | + | + | + | + | + | | | |
| *Billimoria et al. (2013)* | + | + | + | | + | + | + | + | | |
| *DeMauro et al. (2013)* | + | + | + | + | + | + | + | | | |
| *Godfrey et al. (2013)* | + | | + | + | + | | + | | | |
| *Manani et al. (2013)* | + | + | + | + | + | + | + | | | |
| *Castrodale & Rinehart (2014)* | + | | + | | + | | + | | | |
| *Pinheiro et al. (2014)* | + | + | + | | + | | + | | | |
| *Sivanaridan, Sankar & Deorari (2016)* | + | + | + | + | + | | + | | | |
| *Yip et al. (2017)* | + | + | + | + | + | | + | | | |
| *Caldas et al. (2018)* | + | + | + | + | + | | + | | + | + |
| *Bhatt et al. (2020)* | + | + | + | + | + | + | + | + | | + |
| *Croop et al. (2020)* | + | + | + | + | + | + | + | | + | |
| *Dixon et al. (2021)* | + | + | + | | + | + | | | | |
| *Frazer et al. (2021)* | + | + | + | + | + | + | + | + | | |
| *Young et al. (2021)* | + | + | + | + | + | + | + | + | | |
| *Bi et al. (2022)* | + | + | + | + | + | + | + | + | | |
| *Singh et al. (2022)* | + | | + | | + | | + | + | | |

**Note:**
(1) Formation of a multidisciplinary expert team; development of evidence-based interventions; education of hospital staff; (2) Increasing ambient temperature; (3) Using plastic wrap or bag; (4) Regular reinforcements or ongoing process review and feedback loop; (5) Use of radiant warmer or incubator; (6) Using warming mattress; (7) Keeping head in plastic wrap, using polyurethane-lined hats, or stocking-knit caps; (8) Monitoring temperature; (9) Maintaining plastic bag/wrap integrity during the resuscitation process; (10) Delaying anthropometric data collection and non-essential procedures.

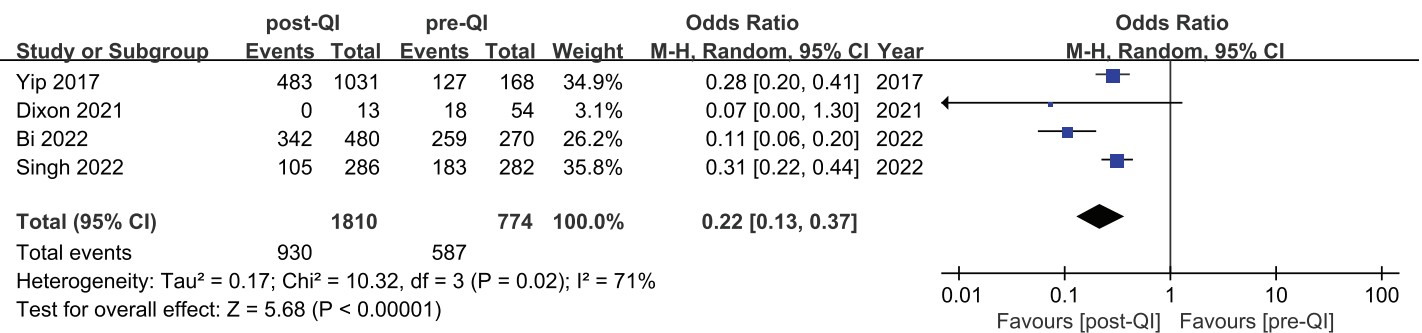

**Figure 2  Forest plot from random effects analysis: The rate of mild hypothermia in VLBW infants pre- and post-QI (*Yip et al., 2017*; *Dixon et al., 2021*; *Bi et al., 2022*; *Singh et al., 2022*).**

Given the substantial heterogeneity between studies ($I^2 = 71\%$), we identified the multicenter study (*Bi et al., 2022*) as a potential source of heterogeneity. Excluding that study led to homogeneity among the studies ($I^2 = 0\%$), and the meta-analysis result remained stable (Appendix 3).

Two studies provided insights into mild hypothermia occurrences among ELBW infants or infants with a GA < 28 w (*Godfrey et al., 2013*; *Singh et al., 2022*). The first study

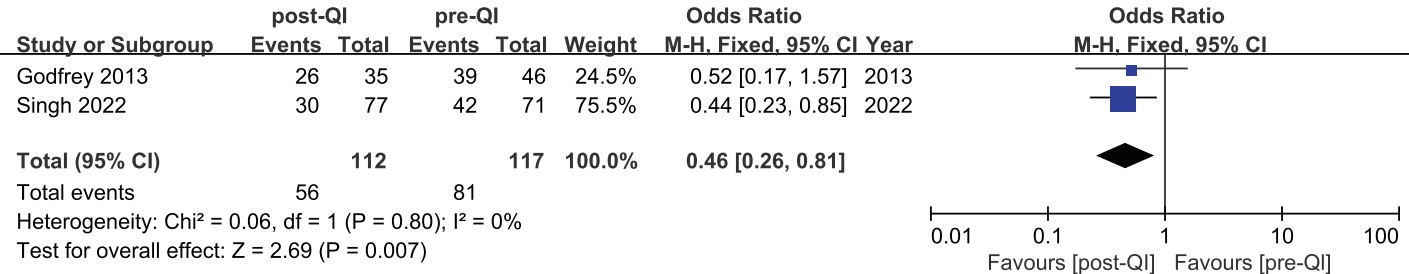

| Study or Subgroup | post-QI Events | Total | pre-QI Events | Total | Weight | Odds Ratio M-H, Fixed, 95% CI | Year |
|---|---|---|---|---|---|---|---|
| Godfrey 2013 | 26 | 35 | 39 | 46 | 24.5% | 0.52 [0.17, 1.57] | 2013 |
| Singh 2022 | 30 | 77 | 42 | 71 | 75.5% | 0.44 [0.23, 0.85] | 2022 |
| Total (95% CI) | | 112 | | 117 | 100.0% | 0.46 [0.26, 0.81] | |
| Total events | 56 | | 81 | | | | |

Heterogeneity: Chi² = 0.06, df = 1 (P = 0.80); I² = 0%
Test for overall effect: Z = 2.69 (P = 0.007)

**Figure 3 Forest plot from fixed effects analysis: The rate of mild hypothermia in ELBW infants pre- and post-QI (*Godfrey et al., 2013*; *Singh et al., 2022*).**

| Study or Subgroup | post-QI Events | Total | pre-QI Events | Total | Weight | Odds Ratio M-H, Fixed, 95% CI | Year |
|---|---|---|---|---|---|---|---|
| Lee 2008 | 54 | 170 | 102 | 134 | 13.1% | 0.15 [0.09, 0.24] | 2008 |
| Manani 2013 | 24 | 230 | 27 | 59 | 6.5% | 0.14 [0.07, 0.27] | 2013 |
| Pinheiro 2014 | 31 | 502 | 56 | 164 | 13.4% | 0.13 [0.08, 0.21] | 2014 |
| Sivanaridan 2016 | 4 | 32 | 12 | 24 | 2.0% | 0.14 [0.04, 0.53] | 2016 |
| Caldas 2018 | 36 | 257 | 81 | 218 | 12.7% | 0.28 [0.18, 0.43] | 2018 |
| Frazer 2021 | 9 | 166 | 43 | 168 | 6.8% | 0.17 [0.08, 0.35] | 2021 |
| Bi 2022 | 144 | 480 | 185 | 270 | 28.0% | 0.20 [0.14, 0.27] | 2022 |
| Singh 2022 | 35 | 286 | 117 | 282 | 17.5% | 0.20 [0.13, 0.30] | 2022 |
| Total (95% CI) | | 2123 | | 1319 | 100.0% | 0.18 [0.15, 0.22] | |
| Total events | 337 | | 623 | | | | |

Heterogeneity: Chi² = 7.35, df = 7 (P = 0.39); I² = 5%
Test for overall effect: Z = 18.93 (P < 0.00001)

**Figure 4 Forest plot from fixed effects analysis: The rate of moderate hypothermia in VLBW infants pre- and post-QI (*Lee, Ho & Rhine, 2008*; *Manani et al., 2013*; *Pinheiro et al., 2014*; *Sivanaridan, Sankar & Deorari, 2016*; *Caldas et al., 2018*; *Frazer et al., 2021*; *Bi et al., 2022*; *Singh et al., 2022*).**

involved 145 cases (*Singh et al., 2022*). It revealed that the incidence of mild hypothermia dropped from 59.1% to 38.9%. Conversely, the second study, conducted at the University of Pittsburgh School of Nursing and Magee-Women's Hospital of the University of Pittsburgh Medical Center in the United States (*Godfrey et al., 2013*), included 81 cases and found no statistically significant change in the incidence of mild hypothermia. A subsequent meta-analysis combining the effect sizes from both studies determined an overall OR = 0.46, 95% CI [0.26–0.81], and I² = 0%, indicating a high degree of homogeneity among the findings (Fig. 3).

## Decrease in moderate hypothermia (body temperature < 36.0 °C)

Among the 12 studies included in the meta-analysis, eight reported on the incidence of moderate hypothermia in VLBW infants or infants with a GA < 32 w (*Bi et al., 2022*; *Caldas et al., 2018*; *Frazer et al., 2021*; *Lee, Ho & Rhine, 2008*; *Manani et al., 2013*; *Pinheiro et al., 2014*; *Singh et al., 2022*; *Sivanaridan, Sankar & Deorari, 2016*). The combined sample size was 3,442, comprising 2,123 participants in the QI intervention group and 1,319 in the control group. Initially, the incidence of moderate hypothermia ranged from 25.5% to

|  | | post-QI | | pre-QI | | | Odds Ratio | | Odds Ratio |
| Study or Subgroup | Events | Total | Events | Total | Weight | M-H, Random, 95% CI | Year | M-H, Random, 95% CI |
|---|---|---|---|---|---|---|---|---|
| Lee 2008 | 21 | 56 | 44 | 48 | 27.9% | 0.05 [0.02, 0.17] | 2008 | |
| Billimoria 2013 | 43 | 76 | 102 | 133 | 37.8% | 0.40 [0.22, 0.73] | 2013 | |
| Singh 2022 | 11 | 77 | 25 | 71 | 34.3% | 0.31 [0.14, 0.68] | 2022 | |
| Total (95% CI) | | 209 | | 252 | 100.0% | 0.21 [0.08, 0.58] | | |
| Total events | 75 | | 171 | | | | | |

Heterogeneity: Tau² = 0.62; Chi² = 9.08, df = 2 (P = 0.01); I² = 78%
Test for overall effect: Z = 3.01 (P = 0.003)

0.01  0.1  1  10  100
Favours [post-QI]   Favours [pre-QI]

**Figure 5** Forest plot from random effects analysis: The rate of moderate hypothermia in ELBW infants pre- and post-QI (*Lee, Ho & Rhine, 2008*; *Billimoria et al., 2013*; *Singh et al., 2022*).

76.1%. Following the introduction of the QI bundle, this rate decreased significantly to between 5.4% and 31.7%. A notable reduction in moderate hypothermia was observed across all studies, with the meta-analysis indicating a combined OR = 0.18, 95% CI [0.15–0.22], and a heterogeneity (I²) of 5% (Fig. 4).

Three studies highlighted the incidence of moderate hypothermia in ELBW infants or infants with a GA < 28 w (*Billimoria et al., 2013*; *Lee, Ho & Rhine, 2008*; *Singh et al., 2022*). These studies included a total of 461 subjects, split between 209 in the QI group and 252 in the control group. The incidence of moderate hypothermia before and after the QI intervention was reported as 76.6% to 56.5%, 35.2% to 14.2%, and 91.6% to 37.5%, respectively. The aggregated OR = 0.21, 95% CI [0.08–0.58], and I² = 78% (Fig. 5). After excluding an early study (*Lee, Ho & Rhine, 2008*), the heterogeneity disappeared (I² = 0%) (Appendix 4). Differences in standards of neonatal care across various temporal phases may contribute to heterogeneity.

## Increase in hyperthermia (body temperature > 37.5 °C)

Six studies assessed the incidence of hyperthermia pre- and post-QI in VLBW infants and preterm infants with GA < 32 w (*Bi et al., 2022*; *Caldas et al., 2018*; *Frazer et al., 2021*; *Manani et al., 2013*; *Singh et al., 2022*; *Yip et al., 2017*). The total sample included 3,615 subjects, with 1,165 cases evaluated before and 2,450 after the interventions. Initially, hyperthermia incidence ranged from 0% to 2.4%, increasing to between 0.4% and 7.8% post-intervention. Among these six studies, two studies (*Caldas et al., 2018*; *Frazer et al., 2021*) highlighted a statistically significant rise in hyperthermia incidence post-intervention, yielding a combined OR = 2.79, 95% CI [1.53–5.09] (I² = 20%). Meta-analytic findings suggest that the implementation of a QI bundle significantly increased the incidence of hyperthermia in this demographic (Fig. 6).

Three studies investigated hyperthermia rates pre- and post-QI in ELBW infants and preterm infants with GA < 28 weeks (*Billimoria et al., 2013*; *Godfrey et al., 2013*; *Singh et al., 2022*), involving a total of 438 participants (250 pre-QI, 188 post-QI). According to Billimoria's study (*Billimoria et al., 2013*), no hyperthermia cases were reported before or after QI. Conversely, the remaining two studies (*Godfrey et al., 2013*; *Singh et al., 2022*) reported hyperthermia incidences of 2.8% and 1.4%, and 2.1% and 5.7%, respectively,

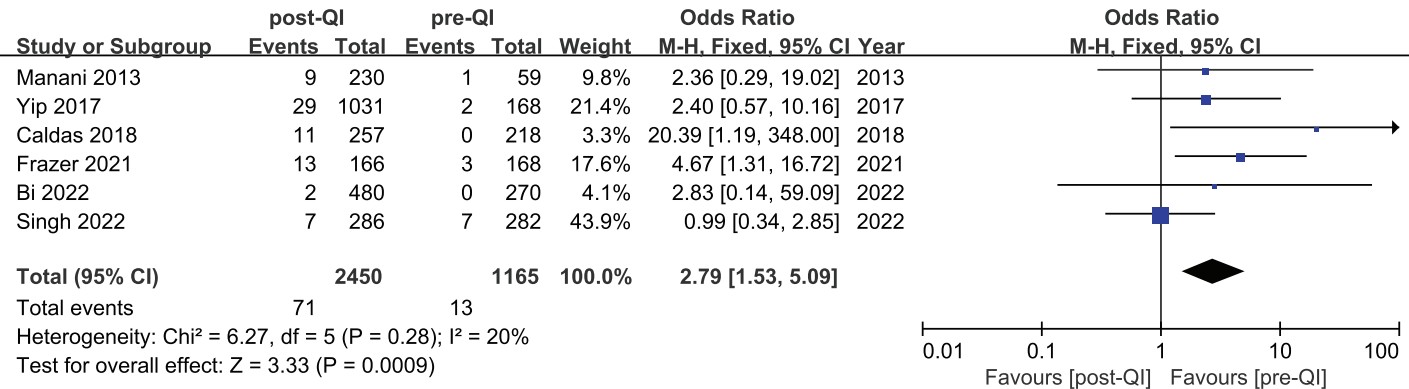

**Figure 6** Forest plot from fixed effects analysis: The rate of hyperthermia in VLBW infants pre- and post-QI (*Manani et al., 2013*; *Yip et al., 2017*; *Caldas et al., 2018*; *Frazer et al., 2021*; *Bi et al., 2022*; *Singh et al., 2022*).

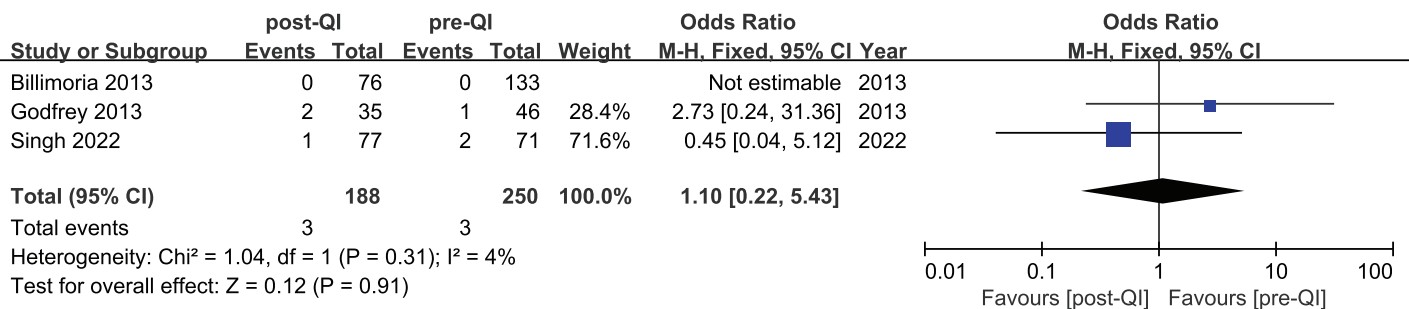

**Figure 7** Forest plot from fixed effects analysis: The rate of hyperthermia in ELBW infants pre- and post-QI (*Billimoria et al., 2013*; *Godfrey et al., 2013*; *Singh et al., 2022*).

before and after QI. Meta-analysis indicated no significant difference in hyperthermia rates post-QI, with a combined OR = 1.10, 95% CI [0.22–5.43], $I^2$ = 4% (Fig. 7).

## Publication bias

Due to the number of studies included for mild hypothermia, moderate hypothermia, and hyperthermia in both VLBW and ELBW groups not exceeding ten—with the maximum being only eight—a funnel plot was not generated to assess publication bias.

## DISCUSSION

Quality improvement is the combined and unceasing efforts of healthcare professionals, patients and their families, researchers, payers, planners and educators to make the changes that will lead to better patient outcomes, better system performance and better professional development (*Batalden & Davidoff, 2007*). This approach has seen widespread application across neonatal medicine (*Alshaikh et al., 2023*; *Bromiker et al., 2024*; *Senaldi et al., 2024*). The results of this systematic review and meta-analysis indicate that QI bundles can significantly decrease the incidence of hypothermia in VLBW/ELBW infants at birth, including mild and moderate hypothermia. Meanwhile, QI bundles may increase the incidence of hyperthermia in VLBW infants, but not in ELBW infants.

In the context of reducing moderate hypothermia (T < 36.0 °C), studies targeting both VLBW and ELBW groups have demonstrated statistically significant reductions in its incidence following QI, with consistent effectiveness across the interventions. However, variability was noted in the baseline rates of moderate hypothermia and the extent of improvement post-QI among the studies. Specifically, pre-QI incidence rates ranged from 25.5% to 76.1% in the VLBW group and 35.2% to 91.6% in the ELBW group, while post-QI rates fell to 5.4%–31.7% and 14.2%–56.5%, respectively. These findings indicate that the quality of care and the implementation process of QI could significantly affect outcomes. Additionally, the diversity in the number and types of interventions within the QI bundles suggests that variations in bundle composition might have influenced the observed differences in outcomes.

Regarding mild hypothermia (T < 36.5 °C), although the results of the meta-analysis for the VLBW and ELBW groups show that the incidence of mild hypothermia significantly decreased after QI, one of the four studies included in the VLBW group showed no statistically significant difference in the incidence of mild hypothermia before and after QI. Similarly, one of the two studies included in the ELBW group also indicated that the reduction in the incidence of mild hypothermia was not statistically significant after QI. This suggests that, compared to moderate hypothermia, the conclusions regarding the improvement in mild hypothermia vary between studies, indicating that it is more challenging to reduce mild hypothermia compared to reducing moderate hypothermia.

Hyperthermia can also lead to adverse outcomes (*Cavallin et al., 2020*; *Zhu et al., 2021*). It is crucial to consider whether QI interventions might increase the incidence of hyperthermia in this population. The meta-analysis results show that there is a statistically significant increase in the incidence of hyperthermia in the VLBW group, whereas the ELBW group did not experience an increase in hyperthermia rates (statistically insignificant). This suggests that the ELBW infants, due to faster heat loss, are less likely to develop hyperthermia with QI bundles. In contrast, VLBW infants have a stronger ability to maintain body temperature, and multi-measure QI bundles might lead to an increase in the incidence of hyperthermia in this group. Therefore, future QI efforts should pay attention to monitoring body temperature and timely adjustment of warming measures to prevent hyperthermia, especially in VLBW infants.

A total of 10 components were included in all the included QI bundles, with each bundle varying in specific components. The smallest bundle contained four, while the largest included nine. Among these components, three were consistently used across all QI studies: formation of a multidisciplinary expert team; development of evidence-based interventions; education of hospital staff; using a plastic wrap or bag; and use of a radiant warmer or incubator. Given the non-RCT design of the QI projects, it is not possible to conclusively determine which specific components are more effective in reducing hypothermia in VLBW/ELBW infants.

In terms of study heterogeneity, the meta-analysis revealed heterogeneity among the four studies included in the mild hypothermia group of ELBW infants. This heterogeneity disappeared after excluding the only multicenter study (*Bi et al., 2022*), suggesting it as the source of heterogeneity. Heterogeneity was also present in the studies included in the

moderate hypothermia group of VLBW infants. This heterogeneity dissipated after the exclusion of an early study (*Lee, Ho & Rhine, 2008*), indicating that differences in neonatal care across different eras may be a source of heterogeneity.

The findings of this meta-analysis indicate that in regions with a high incidence of hypothermia in VLBW/ELBW infants, QI could be implemented to mitigate hypothermia among this population. Practitioners may select various intervention bundles tailored to the unique attributes of their healthcare settings, and subsequently refine the components of these bundles contingent upon their efficacy in diminishing hypothermia rates and their potential to elevate the risk of hyperthermia. Additionally, during the implementation of QI bundles, continuous temperature monitoring is essential. While avoiding hypothermia, attention should also be paid to preventing hyperthermia, as both conditions can cause adverse outcomes.

Strengths and limitations: This meta-analysis is the first to quantitatively evaluate the efficacy and safety of QI bundles in reducing hypothermia among VLBW/ELBW infants at birth. This systematic review comprehensively gathered relevant studies, summarized the components of various QI bundles, and quantitatively assessed the incidence rates of mild hypothermia, moderate hypothermia, and hyperthermia. All included studies were of moderate to high quality. However, the study also has limitations. Firstly, we did not generate funnel plots to assess publication bias due to the number of studies in each group not exceeding 10. However, publication bias may exist, as studies reporting positive changes due to QI interventions are more likely to be published than those showing no improvement. Secondly, given the non-RCT design of the QI projects, it is difficult to determine which specific components within the QI are more effective at reducing hypothermia in this population. Finally, most of the included studies originate from developed countries, with only three studies from developing countries, potentially limiting the generalizability of the results to middle and low-income regions.

## CONCLUSIONS

This systematic review and meta-analysis demonstrated that QI bundles are effective in reducing hypothermia in VLBW/ELBW infants, but may also lead to increased rates of hyperthermia, particularly in VLBW infants. Most included studies, having been conducted in developed countries, limit the applicability of these conclusions to middle- and low-income settings. Furthermore, the design of QI project precludes determining which specific interventions or combinations are most effective in reducing hypothermia in this population. It is advisable to employ QI projects in regions where hypothermia rates among VLBW/ELBW infants are high. Future research should assess the effectiveness and safety of QI bundles in resource-limited settings, explore the most efficacious combinations of QI bundle components, and consider strategies to prevent hyperthermia.

### Funding

This work was supported by the Shenzhen Science and Technology Program Project. (No. JCYJ20220530152414031). The funders had no role in study design, data collection and analysis, decision to publish, or preparation of the manuscript.

### Grant Disclosures

The following grant information was disclosed by the authors:
Shenzhen Science and Technology Program Project: JCYJ20220530152414031.

### Competing Interests

The authors declare that they have no competing interests.

### Author Contributions

- Guichao Zhong analyzed the data, authored or reviewed drafts of the article, and approved the final draft.
- Jie Qi analyzed the data, prepared figures and/or tables, and approved the final draft.
- Lijuan Sheng performed the experiments, prepared figures and/or tables, and approved the final draft.
- Jing Zhuang performed the experiments, prepared figures and/or tables, and approved the final draft.
- Zhangbin Yu conceived and designed the experiments, authored or reviewed drafts of the article, and approved the final draft.
- Benqing Wu conceived and designed the experiments, authored or reviewed drafts of the article, and approved the final draft.

### Data Availability

This is a systematic review/meta-analysis.

### Supplemental Information

Supplemental information for this article can be found online at http://dx.doi.org/10.7717/peerj.18425#supplemental-information.

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
