# Peer review of "Quality improvement bundles to decrease hypothermia in very low/extremely low birth weight infants at birth: a systematic review and meta-analysis"

_PeerJ, doi:10.7717/peerj.18425_

## Round 0.1 · original submission · Minor Revisions

Please address the comments of the reviewers

·

Basic reporting

1.Authors have done the review on an important issue. They have mentioned the limitations of the study
As indicated in the review, all studies did not use same type of temperature support to prevent hypothermia. Regular continuous monitoring of temperature may be stressed since some studies showed hyperthermia also.
2.Over all well written. However, authors can reduce repetetion of contents in the text and Tables.

Experimental design

This is a systematic review with meta-analysis. Methodology adopted appears fine.

Validity of the findings

The findings are valid. Authors may give some suggestions to prevent moderate hypothermia and hyperthermia since they may adversally effect the outcome

Additional comments

Authors may condense the contents especially under results by minimizing repetetion (Tables and text)

·

Basic reporting

• Thank you for an interesting article to review, and an article that is needed to enhance and focus the importance what is done for the Low and extremely low birthweight infants to keep them in normal temperature range.
• The article is clear and unambiguous, using professional English for most parts of the article throughout, but in some suplimentary material there are Chinese characters or use of non english speaking english.
• The literature references, provides a good background/context when used as well as the references in the article. Eventhough I think there are some that are missing from authors such as M. Norman et al. that could have been very benificial for the field understanding and for coverage of a more global understanding on why and how the QI should and or is used.
• The article should include sufficient introduction and background to demonstrate how the work fits into the broader field of knowledge.
• I feel the article is well written and have a professional and easy to follow article structure, with good tablesand figures. And the raw data hat can be shared when a systematic review and meta-analysis is done while data from the original articles is kept silent in a good way.

All Figures are relevant to the content of the article, of sufficient resolution, and appropriately described and labeled.

Experimental design

• Since this is a systematic review and meta-analysis the original primary research is with in the journals within the aims and scope. With is focus on neonates born with Low and extremely low birthweight.
• The research question is well defined, relevant & meaningful. Eventhough the question in it self does not stated how the research fills an identified knowledge gap but rather the conclusion( normal when reviewand meta-analysis studies).
• The research question, is a very relevant and meaningful question and answers many upcomming questions regarding comparisosns between previously published indipendent studies.
• The design is rigorous in the investigation and have been conducted rigorously/done/performed to a high technical & ethical standard.
• The methods section is well described with sufficient details & information to be replicated.

Validity of the findings

• All underlying data have been provided; they are robust, statistically sound, & controlled.

• Conclusions are well stated, linked to original research question & limited to supporting results.

Additional comments

Dear authors and editorial team I have focused also on the following questions in italic and my thoughts after.
a. Is the manuscript clear, relevant for the field and presented in a well-structured manner?
The manuscript is clear and structured, but have missed/ choosen not to display some of similar studies globally that could have been mentioned as references
b. Are the cited references current (mostly within the last 5 years)? Does it include an abnormal number of self-citations?
Yes but some could be newer sicne there are some updates to two of them.
c. Is the manuscript scientifically sound and is the experimental design appropriate to test the hypothesis?
In gereral yes
d. Are the manuscript’s results reproducible based on the details given in the methods section?
Yes
e. Are the figures/tables/images/schemes appropriate? Do they properly show the data? Are they easy to interpret and understand? Are the data interpreted appropriately and consistently throughout the manuscript? Please include details regarding the statistical analysis or data acquired from specific databases.
In my opinion the data and info in tables show a very wide perspective but easy to understand, but an extra figure could say even more.
f. Are the conclusions consistent with the evidence and arguments presented?
Yes
g. Is the conflict of interest funding and acknowledge sections statements adequate.
Yes
h. Is the ethics statements and data availability statements adequate.
yes
i. Are there any gap in knowledge identified?
I belief that the interest and the full article will be very very valuable but a minor revision is needed.

---

## Round 0.2 · accepted · Accept

Thank you for revising your paper based on reviewer feedback, both reviewers now feel the paper is ready for publication based on a re-read of the paper.

·

Basic reporting

Authors have carried out suggested changes

Experimental design

Fine

Validity of the findings

The findings are valid

Additional comments

Since the authors have modified the contents, manuscript may be accepted

·

Basic reporting

The manuscript is much better now.
I still think that a figure would be beneficial for this manuscript, but accept the authors discussion.
Some of the added ref should maybe also be in the tables.

Experimental design

Now better so no comments

Validity of the findings

'no comment', ready now for publication

Additional comments

none